# Forecasting Whole-Brain Neuronal Activity from Volumetric Video

## Abstract

Large-scale neuronal activity recordings with fluorescent calcium indicators are increasingly common, yielding high-resolution 2D or 3D videos. Traditional analysis pipelines reduce this data to 1D traces by segmenting regions of interest, leading to inevitable information loss. Inspired by the success of deep learning on minimally processed data in other domains, we investigate the potential of forecasting neuronal activity directly from volumetric videos. To capture long-range dependencies in high-resolution volumetric whole-brain recordings, we design a model with large receptive fields, which allow it to integrate information from distant regions within the brain. We explore the effects of pre-training and perform extensive model selection, analyzing spatio-temporal trade-offs for generating accurate forecasts. Our model outperforms trace-based forecasting approaches on ZAPBench, a recently proposed benchmark on whole-brain activity prediction in zebrafish, demonstrating the advantages of preserving the spatial structure of neuronal activity.

## 1 Introduction

Recent advances in imaging techniques have enabled the recording of neuronal activity at unprecedented resolution and scale. Light-sheet imaging allows recording of whole-brain activity for small animals, such as the larval zebrafish (Hillman et al., 2019). Raw recordings are in the form of volumetric videos, with hundreds of millions voxels per time step, recorded over hours. Typically, heavy postprocessing is applied to reduce dimensionality of this data down to 1D time traces of activity for distinct regions of interest representing individual neurons or clusters of cells (Abbas & Masip, 2022). Inspired by the success of deep learning models in analyzing minimally processed data in other fields, such as weather and climate forecasting (Rasp et al., 2020; Andrychowicz et al., 2023), we explore the potential of building predictive models directly on such volumetric videos, avoiding any information loss.

The ability to predict future behavior based on past observations is a cornerstone of scientific modeling across a diverse range of domains, ranging from physics to social sciences. Until now, it has not been applied in the context of whole-brain activity in a vertebrate. The recently introduced Zebrafish Activity Prediction Benchmark (ZAPBench) (Anonymous, 2024) aims to change that, taking advantage of datasets that can now be acquired with modern microscopy techniques. ZAPBench provides a rigorous evaluation enabled by the comparison of future brain activity predicted from past brain activity to actual experimental recordings, thereby achieving an objective measure for evaluating predictive models of brain function. The dataset used in ZAPBench is a whole-brain recording from a larval zebrafish, collected using a light-sheet microscopy setup (Vladimirov et al., 2014). The raw volume is made of ∼1.5 trillion voxels, which is reduced in size by three orders of magnitude to a trace matrix of time series by applying a neuron segmentation mask. ZAPBench is the first benchmark that poses the forecasting problem for a significant fraction of neurons in a single brain and provides the raw volumetric recordings for the experiments.

To test the viability of end-to-end forecasting on such data, we propose to use a video model based on techniques that have not been applied to this domain previously. Since processing in the brain is highly distributed (Urai et al., 2022; Naumann et al., 2016), we hypothesize that large receptive fields are important. Furthermore, in comparison to models applied to activity traces, we expect the following advantages. First, by utilizing the entire video as input, a video-based model is not reliant

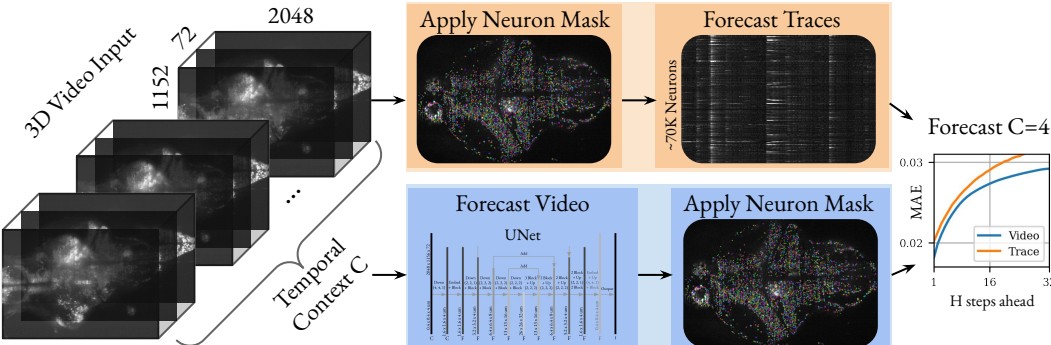

Figure 1: We propose to model light-sheet microscopy recordings of neural activity directly as volumetric video for forecasting instead of extracting and modelling neuron traces. Specifically, we train a model directly on the video and mask the output to optimize the per-neuron mean absolute error (MAE). We find that a UNet performs particularly well for small temporal context and can more effectively utilize spatial contextual information than trace-based time series models.

on the precision of neuron segmentation masks (masks are still applied to predicted frames for direct comparison to trace-based methods). Second, the inherent grid structure of the video preserves the spatial relationships between neurons, information that is otherwise lost during trace extraction. Finally, such a model can leverage potentially relevant visual cues present in the voxels between segmented cells or within the voxels of individual cell masks to enhance forecasting accuracy.

Building a model for this problem poses fundamental engineering challenges. Standard video models operate on 2D frames, and the presence of the additional z dimension naturally complicates scaling. In ZAPBench, a single XY slice of a 3D frame has a native size of $2048 \times 1328$ pixels, and is thus comparable to a frame of a natural 1080p video. Every 3D frame is composed of 72 such slices, increasing the volume of the input data by up to two orders of magnitude relative to such videos and resulting in several hundreds of megabytes per frame.

For our model, we choose a variant of the UNet (Ronneberger et al., 2015) and adapt it to 4D data. We also develop a data input pipeline where both input and model are spatially sharded across multiple hosts and accelerators. To maintain a manageable size of the intermediate activations, we represent temporal input frames as channels. This approach allows us to explore the impact of varying spatial context by manipulating the receptive field while keeping computational cost (FLOPS) roughly constant.

We perform extensive experiments to construct an effective video model for neuronal activity forecasting on ZAPBench. Despite the success of pre-training in other domains (Devlin, 2018; Bao et al., 2021), we find it not to be a useful technique for improving forecast accuracy, even when using an order of magnitude more data recorded from other specimens of the same species. Further, we investigate the effect of input resolution, spatial context, and temporal context of the model on the forecast accuracy. Surprisingly, we find that lowering input resolution by up to 4x can be beneficial for performance and observe a clear trade-off between spatial and temporal context.

Our models, which implicitly capture the spatial relationships within their field of view, can improve forecast accuracy beyond that achieved by trace-based models on ZAPBench, especially when only short temporal context is available. On ZAPBench, multivariate trace-based models, which can in principle learn functional relationships between cells, do not perform significantly better than univariate models that treat all cells independently and identically. Our proposed model is therefore the only multivariate model that can consistently outperform univariate models on this benchmark.[1]

In summary, our contributions are as follows:

1. We propose to forecast zebrafish neuronal activity recorded using light-sheet microscopy directly in the native domain as volumetric video (3D + time).

---

[1]Comparisons between video and trace-based models are also included in ZAPBench (Anonymous, 2024).

2. We empirically show that the input resolution and pre-training on additional volumetric videos from similar specimens have negligible impact on the results.

3. We perform exhaustive model selection to quantify the impact of spatial (XYZ) and temporal context size for activity forecasting accuracy and find a clear trade-off.

4. On ZAPBench, our proposed model is the only approach that consistently benefits from multivariate information, and therefore achieves leading performance for short temporal context.

## 2 FORECASTING NEURONAL ACTIVITY FROM VIDEO

We propose to forecast neuronal activity in the ZAPBench dataset (Anonymous, 2024) directly in the volumetric video domain. Specifically, we utilize a temporal context of $C$ video frames to predict the subsequent $H$ frames. Per-neuron forecasts and loss are then computed by applying the segmentation mask to the predicted video frames. This contrasts with the traditional approach, which applies the segmentation mask to the original video data to extract activity traces before performing any forecasting. See Figure 1 for a comparison of these two approaches.

The ZAPBench dataset comprises high-resolution, whole-brain activity recordings of a larval zebrafish engaged in various behavioral tasks. Data was acquired using light-sheet fluorescence microscopy, enabling real-time imaging of neuronal activity at cellular resolution. This was made possible by using an animal genetically modified to express GCaMP (Dana et al., 2019), a fluorescent calcium sensor, in the nuclei of its neurons. ZAPBench provides both preprocessed activity traces for approximately 70,000 neurons and the corresponding raw volumetric video data. This raw data, denoted as $\mathbf{Y}$, has dimensions of $2048 \times 1152 \times 72 \times 7879$ (XYZT) and a resolution of $406\,\mathrm{nm} \times 406\,\mathrm{nm} \times 4\,\mu\mathrm{m} \times 914\,\mathrm{ms}$. We use a center crop of $1328$ voxels in Y due to negligible cell activity in the border regions. Models forecasting $H = 32$ time steps are benchmarked using short ($C = 4$) or long ($C = 256$) temporal context.

Anonymous (2024) preprocess the raw volumetric video by aligning each frame to a reference volume for stabilization so that the neuron segmentation masks can be statically applied throughout the experiment. Further, a standard "$\Delta F/F$" normalization scheme is applied to the voxel intensities, with $F$ denoting a baseline value (Mu et al., 2019; Zhang et al., 2023). The normalized signal is in the $[-0.25, 1.5]$ range.

The neuron segmentation model is specifically trained for the dataset and yields 71,721 neurons. Formally, the segmentation mask can be considered as a mapping $\mathrm{seg}\colon \mathcal{N} \to 2^{\mathcal{S}}$ from integer identity of a neuron $\mathcal{N} = [71721]$ to a set of three-dimensional spatial indices, which is an element of the power set of index locations $\mathcal{S} = [2048] \times [1328] \times [72]$. The neuron activity at an arbitrary timestep $t$ is then given by *averaging* the activity over spatial locations associated with each cell, i.e.:

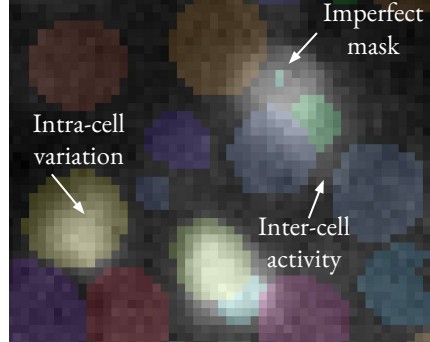

$$\mathrm{y}_n(t) = \frac{1}{|\mathrm{seg}(n)|} \sum_{s \in \mathrm{seg}(n)} \mathbf{Y}_s(t). \qquad (1)$$

While this is a natural choice, it loses information related to cell size, position and spatial distribution of intensities within it, and completely discards voxels that are not part of any segmentation mask or incorrectly segmented. Figure 2 depicts these potential issues.

Figure 2: Illustration of potential loss of information when segmenting neurons. The colored objects are predicted segmentation masks. A fragment of a 2d slice of the activity video is shown in greyscale.

We instead apply a video model to the raw input frames and directly forecast volumetric frames while optimizing and measuring the mean absolute error (MAE) on the segmented neurons. Prior work in neural response prediction (Schoppe et al., 2016; Cadena et al., 2019) has proposed additional metrics that explicitly take trial-to-trial variability into account. The experimental setting used in ZAPBench did not allow for sufficiently numerous trial repetitions to make these metrics applicable, but we note them as an interesting direction to explore in future work if

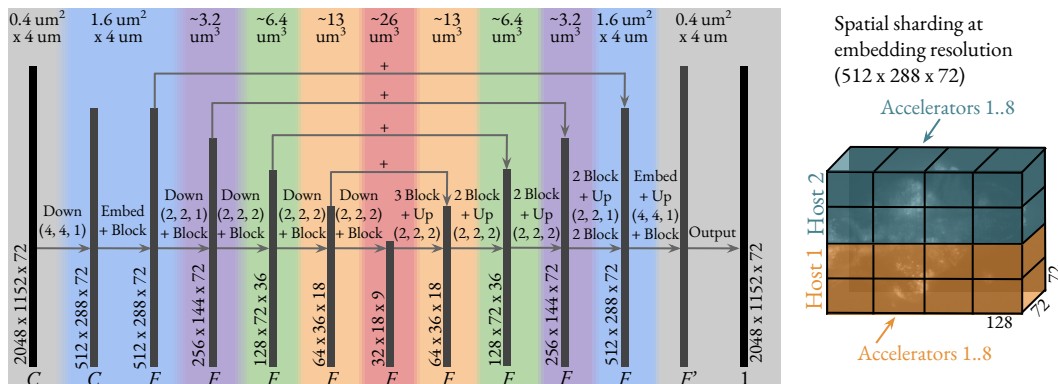

Figure 3: Architecture and input sharding overview. **A**: We use a variation of the UNet architecture (Ronneberger et al., 2015) with 3D spatial input and treat the $C$ input frames as channels. Further, we use a fixed number of features at every resolution to improve scalability. The network is conditioned on the time horizon $H$ and outputs a single volumetric frame at a time, similar to MetNet-3 (Andrychowicz et al., 2023). To control for spatial context at constant FLOPS, four blocks at the lowest resolution can be replaced by one block of higher resolution. **B**: Data loading and the network are spatially sharded and allow for flexible scaling to full resolution inputs.

calcium recordings made with increased number of trials become available. In addition to MAE, we also report correlations between the predicted and actual activity in App. A.4.

One frame of the volumetric video can be described as $\mathbf{Y}(t) \in \mathcal{V}$ with $\mathcal{V} = \mathbb{R}^{2024} \times \mathbb{R}^{1152} \times \mathbb{R}^{72}$. A video model with a $P$-dimensional weight vector $\mathbf{w} \in \mathbb{R}^{P}$ can then be denoted as $\mathbf{f} : \mathcal{V}^{C} \times \mathbb{R}^{P} \rightarrow \mathcal{V}^{H}$. That means the video model $\mathbf{f}$ receives a 4D volumetric input with $C$ frames, outputs $H$ frames, and is parameterized with weights $\mathbf{w}$. We obtain the prediction of the $h$-th frame as

$$\hat{\mathbf{Y}}(t, h) = \mathbf{f}_h(\mathbf{Y}(t), \dots, \mathbf{Y}(t + C), \mathbf{w}), \tag{2}$$

and denote by $\hat{\mathbf{y}}(t, h)$ the corresponding 1D trace vector computed using Eq. 1. For a fair comparison with trace-based models in ZAPBench we optimize the trace-based MAE $\mathcal{L}$ over all training timesteps $T_{\text{train}}$ with respect to the model parameters $\mathbf{w}$

$$\mathcal{L}(\mathbf{w}) = \frac{1}{|T_{\text{train}}|\, H\, |\mathcal{N}|} \sum_{t \in T_{\text{train}}} \sum_{h \in [H]} \sum_{n \in \mathcal{N}} |\mathbf{y}_n(t + h) - \hat{\mathbf{y}}_n(t, h)|\,. \tag{3}$$

If we instead optimize the voxel-wise MAE, the models perform relatively worse when evaluated on the trace-based MAE because it corresponds to a different weighting of neurons by their size in number of voxels.

## 3 SCALABLE VOLUMETRIC VIDEO ARCHITECTURE

Efficiently training models consuming high-resolution volumetric video of varying input context sizes $C$ requires a scalable architecture and data loading system. We achieve this by extending a standard UNet architecture (Ronneberger et al., 2015) to 4D by mapping temporal input context to features of the first convolutional layer, conditioning on lead-time to predict only single frames, and sharding both the model and the data loading process. Figure 3 shows the intermediate resolutions and representation sizes. The network comprises a series of pre-activation residual convolutional blocks (He et al., 2016) with fixed feature size $F = 128$, each with two group normalization layers (Wu & He, 2018) using 16 groups, Swish activation (Ramachandran et al., 2017), and $3^3$ convolutions for XYZ throughout.

### 3.1 TEMPORAL INPUT CONTEXT AS FEATURES

Typically, video UNet variations use color channels as input features (Gao et al., 2022; Ho et al., 2022a) and convolve over frames using a temporal convolution (Ho et al., 2022b). This approach

is intractable in our case because of the additional Z dimension. Instead, we treat the temporal input context of $C$ frames as input features to the UNet. This confers the following advantages: 1) the temporal sizes of the input and output are decoupled, 2) the network parameter count is easily controlled, 3) representation sizes and computation requirements are reduced while using more features, and 4) early layers of the network have access to long-range temporal dependencies. Our model is similar to architectures used in standard time series models, which often treat temporal context as features (Zeng et al., 2023; Chen et al., 2023).

### 3.2 Varying the Receptive Field

We design a flexible UNet architecture that can adapt the receptive field while keeping the computational cost (FLOPS) fixed. We find that full native resolution is not necessary for optimal prediction accuracy (see Sec. 4.1), and thus downsample the input by a factor of 4 in XY using averaging. The first resampling block then uses a factor 2 in XY to achieve roughly isotropic resolution in XYZ, while the following ones downsample equally in all dimensions. We always use four residual blocks at the lowest resolution, and three at all other resolutions. This allows us to change the receptive field while keeping the FLOPS roughly fixed by removing the four lowest resolution blocks and instead adding one block to the respective next higher resolution. This is because one block at the higher resolution requires as many FLOPS as four blocks after downsampling by a factor of two in X and Y. In an ablation, we show that controlling for FLOPS is sensible because increasing the parameter count does not increase performance further (see Figure 7).

The receptive field along a dimension depends on the cumulative product of the downsampling factors and the number of convolutions at the lowest resolution,

$$\texttt{receptive\_field}_{\texttt{dim}} = \texttt{cum\_downsampling\_factor}_{\texttt{dim}} \times \texttt{num\_blocks} \times 4,$$

where the factor 4 is because every block has two convolutions, each of which increase the receptive field by two. For a network that does not downsample at all, as for example used in Sec. 4.1.1, to account for the input and output convolutions and the center voxel, we have to increase the receptive field size by five. Therefore, the architecture depicted in Figure 3 has a receptive field of $(1024, 1024, 128)$ in XYZ comparable to the size of the complete frame. We tried to further enhance the receptive field to cover the whole frame using a multi-axis vision transformer (Tu et al., 2022) at the lowest resolution, but did not observe any accuracy gains. For the output, we upsample twice to obtain the original resolution, and use one residual block per resolution, but with a reduced feature dimension of $F' = 32$ to keep hidden representations at a manageable size.

### 3.3 Lead-time Conditioning

Instead of forecasting autoregressively or predicting the complete horizon of $H$ frames in a one-shot way, we condition the network on an integer lead-time $h \in [H]$ and predict the corresponding single frame independently as proposed by Andrychowicz et al. (2023) for weather forecasting. During data loading, we sample a lead time and the corresponding target frame uniformly at random from $[H]$. This requires loading only a single target frame per sample and, in line with previous results from weather forecasting (Rasp et al., 2020), performs better than frame-level autoregressive prediction on our problem. Every convolutional block in the network is conditioned using a FiLM layer (Perez et al., 2018) on the lead time encoded using a 32-dimensional sinusoidal embedding (Vaswani et al., 2017). Figure 4 shows that directly predicting all $H$ frames tends to overfit while lead-time conditioning performs equally well with both MAE and HL-Gauss (Farebrother et al., 2024), a distributional regression objective that results in slightly faster model convergence. However, in our experiments we use the conditioned MAE for its simplicity and because it does not require binning, which might complicate pre-training on datasets of slightly different scale.

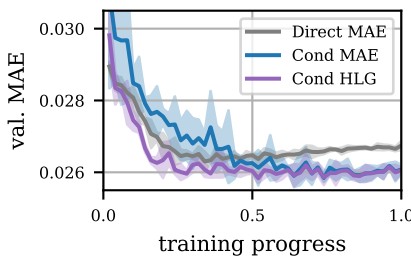

Figure 4: Comparison of direct MAE and lead-time conditioned variants.

### 3.4 SHARDED DATA LOADING AND MODEL

Despite the scalability features of the proposed UNet model for volumetric video, in practice applying it requires distributing the input and hidden representations across accelerators and machines. We train all models in Sec. 4 using a single sample per batch, noting that this can already correspond to several GBs of input data. We use spatial sharding in XY using the `jax.Array` API (Bradbury et al., 2018) so that each box in Figure 3B is handled by an individual accelerator. We also implement a custom data loader that distributes data loading across hosts so that each machine only loads the necessary boxes. To achieve this, we chunk our data in the `zarr3` format (Miles et al., 2023) and use the TensorStore API (TensorStore developers, 2024) to load and collate chunks. Our data loader follows the jax sharding automatically.

## 4 EXPERIMENTAL RESULTS

We present experimental results evaluating the proposed volumetric video model on ZAPBench, a benchmark for whole-brain neuronal activity prediction for a larval zebrafish (Anonymous, 2024). Uniquely, ZAPBench provides the raw volumetric recordings for most of the neurons in the brain enabling data-driven approaches like ours. First, in Sec. 4.1 we empirically select and validate the final architecture variant used for the benchmark. In particular, we investigate the trade-off between temporal context $C$ and spatial context in the form of the receptive field to assess the need for multivariate models. Further, we evaluate the feasibility of pre-training on additional zebrafish specimens as well as the effect of input resolution. We identify the model depicted in Figure 3 as a strong model for the short context size $C = 4$, where we achieve the best performance across the benchmark, as presented in Sec. 4.2. For the long temporal context $C = 256$, we only see an improvement of forecast accuracy in specific cases.

**Hyperparameters.** Unless stated otherwise, we train every model for 250k to 500k steps by optimizing the trace-based MAE with a batch size of 1 using the AdamW optimizer (Loshchilov et al., 2017) using an initial learning rate of $10^{-4}$ decayed using a cosine schedule (Loshchilov & Hutter, 2017) to $10^{-7}$ and a weight decay factor of $10^{-5}$. Due to their tendency to overfit, we use a dropout rate of 0.1 on the features for long-context models with $C = 256$. These hyperparameters were optimized on the validation set during development. We choose checkpoints based on the validation performance monitored during training. We present experimental results in terms of mean performance and report two standard errors over at least three random seeds that control data loading and parameter initialization. The only exception to this are the high resolution results presented in Sec. 4.1.3, where we only report a single result because of their compute requirements. Most individual training experiments use 16 A100 40GB GPUs.

### 4.1 MODEL SELECTION

We compare between different methods and models to improve performance on the ZAPBench benchmark. For Sec. 4.1.1 and 4.1.2, we downsample the volumetric frames by a factor of four in XY using averaging to $512 \times 288 \times 72$. The segmentation mask is downsampled to the same shape using striding. We investigate the effect of spatial and temporal context and the potential for pre-training on related datasets. In Sec. 4.1.3, we use the full resolution ZAPBench targets and segmentation, and assess the importance of input resolution on performance.

#### 4.1.1 SPATIAL VS. TEMPORAL CONTEXT

We use UNets with different numbers of downsampling blocks to vary the spatial context but keep the FLOPS fixed (see Sec. 3), and find that there is a trade-off between the spatial ($S$) and temporal ($C$) context. We compare models without any downsampling blocks, with two downsampling blocks, and with four. The models have spatial contexts $S$ of 21, 64, and 256 in XY, respectively. Details on the architecture and computation of the receptive field can be found in App. A.1.1. Also note that the spatial context at full resolution of these models would be $4\times$ higher. Figure 5 shows that a short temporal context requires larger spatial context to obtain optimal performance. For long temporal context, however, the models with large spatial context start to overfit and underperform. The effect becomes apparent between a temporal context of 16 and 64. This result suggests that

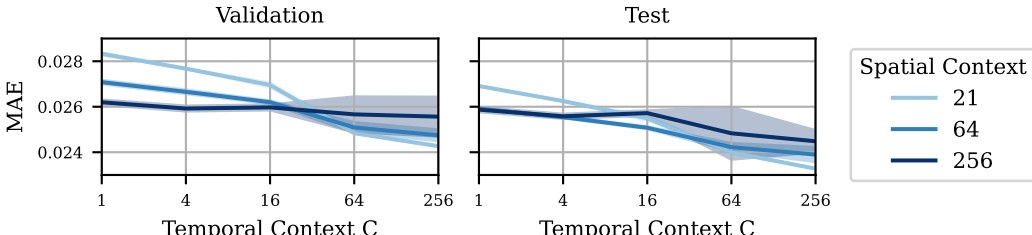

Figure 5: Validation and test performance for varying temporal context sizes $C$ as well as spatial context sizes $S$ with networks having comparable FLOPS. We find that there is a trade-off between spatial and temporal context with a cross-over point between $C = 16$ and $C = 64$, where spatial context stops being useful and leads to overfitting. The periodicity of many conditions is roughly 64, which might explain spatial context becoming redundant. We report the mean and two standard errors.

video models are able to exploit multivariate information for short temporal context but provide little benefit for long context, where univariate models perform equally well (Anonymous, 2024).

### 4.1.2 PRE-TRAINING ON OTHER SPECIMENS

We attempt to pre-train a model on other specimens recorded and preprocessed in a similar way to the zebrafish used for ZAPBench. We pre-train the model either on two additional specimens recorded in the same experimental session, or on these two and six more from two other sessions. Because there is no segmentation available for the other specimens, the model is pre-trained using voxel-based MAE for 800k steps, and then fine-tuned on the ZAPBench dataset for 200k steps using the trace-based MAE. We use three different learning rates, $10^{-4}$, $10^{-5}$, and $10^{-7}$, for fine-tuning, and select the best model by validation performance,

| SETTING | TEST MAE |
|---|---|
| Train | $0.02573 \pm 0.00005$ |
| Pre-train +2 | $0.02590 \pm 0.00005$ |
| Pre-train +8 | $0.02591 \pm 0.00001$ |
| Train + Val | $0.02534 \pm 0.00010$ |

Table 1: Training on more data from the same specimen ("Train + Val") improves performance more than pre-training and finetuning on others. Results shown for $C = 4$ and data $4\times$ downsampled in XY.

which was obtained by fine-tuning with the lowest learning rate. Table 1 shows that pre-training with fine-tuning does not improve performance over standard training. However, training on ∼14% more data from the same specimen does improve performance significantly. Confidence intervals shown are calculated as two standard errors.

### 4.1.3 EFFECT OF INPUT RESOLUTION

We assess the relevance of input resolution when forecasting neuronal activity, and find that, surprisingly, predicting from a lower resolution performs best. We compare three variants of our model: a model that predicts from data $4\times$ downsampled in XY, as depicted in Figure 3, one that downsamples only by factor 2, and one that is parameterized at full native resolution. Therefore, the full-resolution model loads and processes $16\times$ more data. We achieve almost perfectly linear scaling by using proportionally more compute resources, maintaining the same throughput thanks to the sharded data input pipeline and model (see Sec. 3). In all cases, we scale the field of view of the network so that its size in physical units remains constant between experiments (see App. A.1.3).

For $C = 4$, Table 2 shows that the model with the lowest input resolution obtains a trace-based test MAE that is statistically identical with that of the model using intermediate resolution inputs. However, the full resolution model performs significantly worse. This suggests that despite the short temporal context input resolution does not play a major role in improving performance, and that the intracellular voxel-to-voxel variations in the recorded images do not carry information useful for

| INPUT | TEST MAE |
|---|---|
| Downsample $4\times$ | $0.0267 \pm 0.0002$ |
| Downsample $2\times$ | $0.0268$ |
| Full resolution | $0.0273$ |

Table 2: Increasing input resolution does not improve performance, and decreases it slightly at full resolution.

forecasting, which might have applications to the de-
sign of future zebrafish activity recording experiments. We suspect that the decreased performance of the full resolution model could be caused by the significantly increased input voxel-to-parameter ratio while keeping the number of training examples fixed.

## 4.2 PERFORMANCE ON ZAPBENCH

We evaluate the best-performing architectures on ZAPBench for both short and long context settings. In Figure 6, we report the trace-based MAE versus forecasting steps in comparison to the best-performing trace-based models (Anonymous, 2024). We average performance across test sets of different stimulus conditions. For trace-based models, TSMixer (Chen et al., 2023) achieves best performance for short context, $C = 4$, and a univariate MLP for long context, $C = 256$. We use the video-based architecture depicted in Figure 3 for the short context that has a spatial context of $1024 \times 1024 \times 72$ in XYZ, which is global except in X where it covers half the voxels. For the long temporal context, we use a model that does not downsample further than $(4, 4, 1)$ at the input, which we found in Sec. 4.1.1 to work best for this case. This model has a spatial context of $64 \times 64 \times 21$, which corresponds to $26\,\mu\text{m} \times 26\,\mu\text{m} \times 84\,\mu\text{m}$ in XYZ.

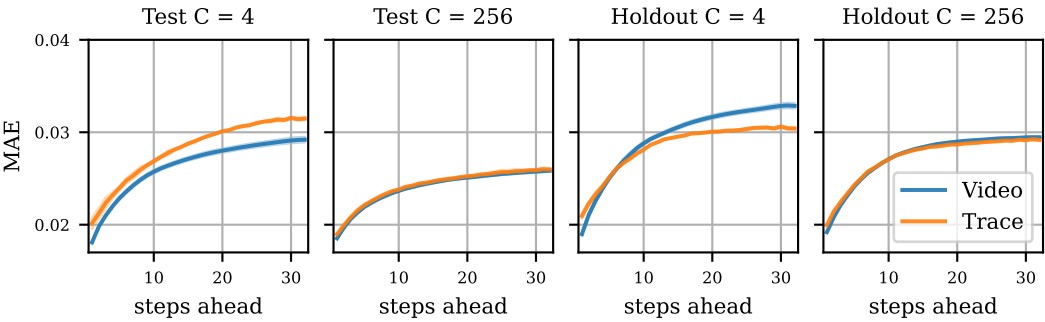

Figure 6: Comparison of volumetric video model with best-performing trace-based model for short (left) and long (right) context on the benchmark test set (averaged over eight conditions) and the experimental condition held out from the training data. We report the mean and two standard errors.

We find that the volumetric video models achieve the best performance in the short context $C = 4$ setting. For $C = 256$, there is no significant difference between the univariate trace-based model and the video model on the test set when evaluated with MAE, but the video model does improve correlation metrics (App. A.4). This aligns with our observation in Sec. 4.1.1, where longer temporal context requires less spatial context for the same forecast accuracy. ZAPBench also holds out one stimulus condition entirely from training. We find that video models generalize better on this holdout condition for one-step-ahead forecasts but not for longer horizons. In App. A.3, we further show model performance separately for each experimental condition. For the short context, we find that the video model performs better in six, equally well in one, and worse in two out of the nine conditions.

More precisely, when evaluated with a context length $C = 4$ on both the test and holdout sets, the video model demonstrates a significant improvement in one-step-ahead forecasting accuracy, achieving about 10 percentage point reduction in error compared to the best performing trace-based model. With $C = 256$, the video model exhibits marginally superior performance in the first few forecasting steps, achieving up to a 2 percentage point reduction in MAE at the first step. Beyond the initial steps, both models demonstrate comparable accuracy on the test set.

What explains the improved performance of the video model relative to the trace-based approaches? In App. A.2 we report results of an experiment in which we masked out all unsegmented voxels, which did not reduce the grand test-MAE. This suggests that segmentation quality is not a significant limitation, that no significant information is contained in the unsegmented regions of the dataset, and that he accuracy gains can be attributed to the better utilization of the spatial distribution of the recorded fluorescence signal. Furthermore, the results in Table 2 and Figure 5 suggest that it

is specifically the correlations between cells in the recorded fluorescence signals, rather than the distribution of signals within individual cells, that drives these improvements.

## 5 CONCLUSION

We presented an approach for forecasting of zebrafish neuronal activity based on utilizing the raw neural recording data as a volumetric video to make predictions. We find that this method has several advantages over traditional trace-based methods. In particular, video-based prediction leverages spatial relationships between neurons that are hard to exploit when reducing the data to 1D traces. This allows for more accurate predictions, especially when working with short temporal contexts. This advance comes at the expense of a significant increase in computational cost (2-3 orders of magnitude relative to trace-based models, see App. A.5 for details).

We report several findings that were contrary to our expectations. First, we surprisingly find that using higher resolution input frames does not improve performance. Second, the commonly used paradigm of pre-training on a larger data set and fine-tuning only leads to reduced forecast accuracy in our experiments. In contrast, we observe that more data from *the same specimen* does improve performance, so we hypothesize that pre-training may be complicated by distribution shifts between specimens, such as differences in signal and noise levels. Finally, increasing model capacity does not always translate to performance improvements but instead leads to overfitting for long temporal context.

Future work could explore the use of probabilistic models, latent space representations, and more sophisticated regularization methods and input augmentations to further improve the accuracy of video forecasting for neuronal activity.

## REPRODUCIBILIY STATEMENT

All relevant code and interactive visualizations of the predictions will be made publicly available following double blind review.

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

# A APPENDIX

## A.1 ARCHITECTURAL DETAILS

Every network has an embedding $3^3$ convolutional layer mapping from temporal context $C$ to $F$ features, and an output convolutional layer mapping from $F'$ (when upsampling) or $F$ to 1 feature, giving the single lead-time conditioned forecast frame. In the downsampling pathway, we apply one convolutional block at every resolution. During symmetric upsampling, we use three convolutional blocks at the lowest resolution, and two for all higher resolutions. Before upsampling to super-resolution (i.e., resolution that is higher than that of the input), we use a convolution to map from $F$ to $F'$ features to reduce the size of intermediate representations. During super-resolution upsampling we use one convolutional block per resolution. Each convolutional block has a pre-activation residual design with the following chained layers: group normalization, swish activation, $3^3$ convolution, group normalization, conditioning on lead time using a FiLM layer, swish activation, optional feature dropout (only used with rate $0.1$ for $C = 256$), and lastly the second $3^3$ convolution. The UNet-structure is realized by adding the representations obtained during sequential downsampling to the upsampled representation. The number of features at every resolution is fixed to $F = 128$, except for the super-resolution upsampling, where it is $F' = 32$.

### A.1.1 SPATIAL VS. TEMPORAL MODELS

This study employs three distinct models based on the aforementioned design.

The first model, maintaining a consistent spatial dimension of $512 \times 288 \times 72$, forgoes downsampling and upsampling blocks. It incorporates four processing blocks at this resolution, along with two convolutional layers at the input and output stages. The receptive field, calculated as $S = 1 + (4 \times 2 + 2) \times 2 = 21$, is determined by considering the central voxel and adding 2 for each $3^3$ convolution.

The second model, downsamples the input data to $64 \times 64 \times 32$ and has a receptive field of $S = 64$. This is derived from the cumulative downsampling factors of $(4, 4, 2)$ in the X, Y, and Z dimensions, respectively, and applying Equation 4.

Similarly, the third model employs downsampling factors of $(16, 16, 8)$, resulting in a $256 \times 256 \times 128$ receptive field. This translates to a global receptive field along the Z-axis, a near-global receptive field along the Y-axis, and a receptive field encompassing half of the total extent along the X-axis.

### A.1.2 LEAD-TIME CONDITIONING

For the results shown in Figure 4, we use three different losses: direct MAE, conditioned MAE, and conditioned HL-Gauss. Apart from the FiLM layers to condition on lead-time, the architecture is the same in all cases, with the exception of the last layer which maps from $F$ to the output dimensionality. The output dimensionality for the direct MAE is the number of forecast timesteps $H$. For the conditioned MAE, it is simply 1, as also described in Figure 3. For the conditioned HL-Gauss loss, it is 32, which is equal to $H$, and each output corresponds to a discretized bin of the data range. The HL-Gauss loss transforms a real value by representing it as a weighted average of bin mean-values, for details see (Farebrother et al., 2024).

### A.1.3 MODELS FOR DIFFERENT INPUT RESOLUTIONS

To investigate the influence of input resolution on model performance, we conducted a comparative analysis. We compared our primary model, which operates on data downsampled by a factor of 4 in the XY plane, with two alternative configurations: one employing a downsampling factor of 2, and another utilizing full-resolution input.

To ensure equitable comparison, the architectures of these models were kept broadly consistent, with necessary adjustments to accommodate the differing input resolutions, while maintaining a consistent full-resolution output frame. Specifically, for the model operating on $2\times$ downsampled input, we augmented the architecture with three additional blocks at the input resolution and removed one upsampling block, relative to the architecture depicted in Figure 3. In contrast, the model utilizing

full-resolution input incorporated two initial downsampling blocks with factors $(2, 2, 1)$ and omitted any super-resolution components, resulting in a conventional UNet architecture.

## A.2 IMPACT OF UNSEGMENTED VOXELS

In contrast to video models which analyze all voxels of the calcium movie, the trace extraction process ignores voxels that do not correspond to segmented cells. This potentially discards information that could be useful in forecasting. To test to which degree this is indeed the case, we trained the $C = 4$ video forecasting model with the unsegmented voxels set to constant value (0). The grand average test MAE for that model ($0.02663 \pm 0.00003$) was not significantly different from that of the video model processing the complete volume ($0.02672 \pm 0.00010$). This indicates that the unsegmented voxels are unlikely to contain information that could improve forecasts and that any gains relative to the trace-based models can be attributed to the utilization of the spatial distribution of the underlying calcium signals within the segmented cells.

## A.3 ADDITIONAL EXPERIMENTAL RESULTS

In Figure 8 and 9, we show a fine-grained version of the benchmark of the trace and video-based models in the main text (Figure 6). The figures show the performance per experimental condition the fish was exposed to. For more details on these conditions, we refer to ZAPBench (Anonymous, 2024). For short context, $C = 4$, we observe that the video-based model performs better on six experimental conditions, and worse for many steps ahead on the "dots", "taxis", and "open loop" conditions. For long context, the video-based model performs almost identical. As in the main text, we display two standard errors about the mean in with shaded regions.

Figure 10 reports performance relative to four trace-based models included in ZAPBench. Figure 11 illustrates MAE differences for a few example frames.

On the right in Figure 7, we further show an ablation to confirm that the improvement of multivariate video models is due to increased receptive field and not because of using more parameters. In particular, in our experimental setup in Sec. 4.1.1 we keep FLOPS fixed instead of number of parameters. In the example on the right, we instead increase the width by a factor of two leading to an incrase in FLOPS by a factor of $4$ while keeping the receptive field fixed. We observe that increasing FLOPS at the same spatial context leads statistically to the same performance. Therefore, the performance improvement observed in Figure 5 is likely due to the increased receptive field, especially for short context. The example on the right is for the case of $C = 4$.

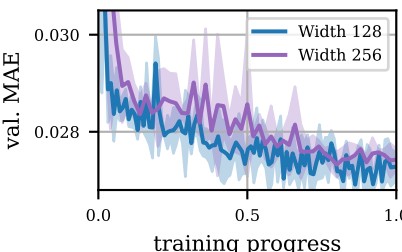

Figure 7: Ablation on increasing parameter count instead of receptive field.

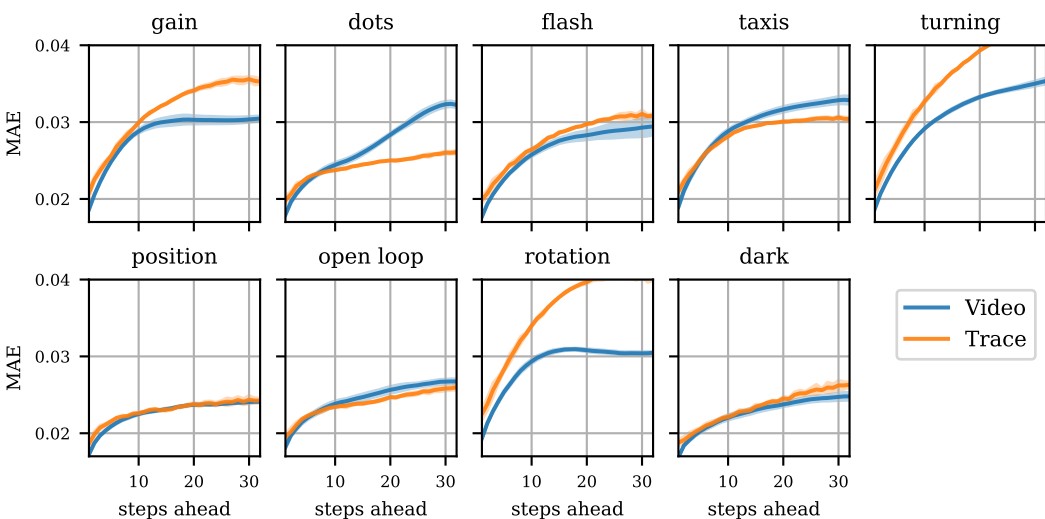

Figure 8: Comparison of volumetric video to best trace-based model on all conditions **for short context,** $C = 4$.

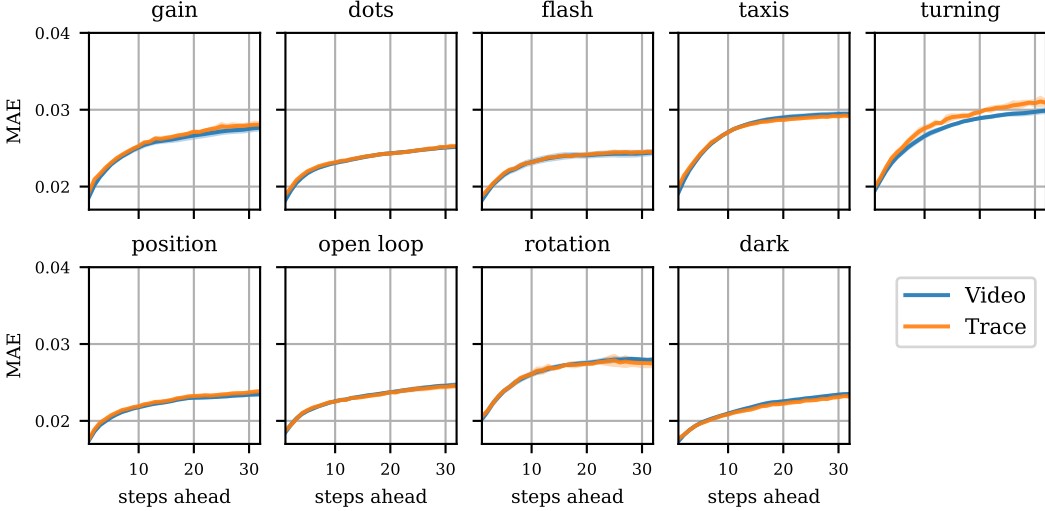

Figure 9: Comparison of volumetric video to best trace-based model on all conditions **for long context,** $C = 256$.

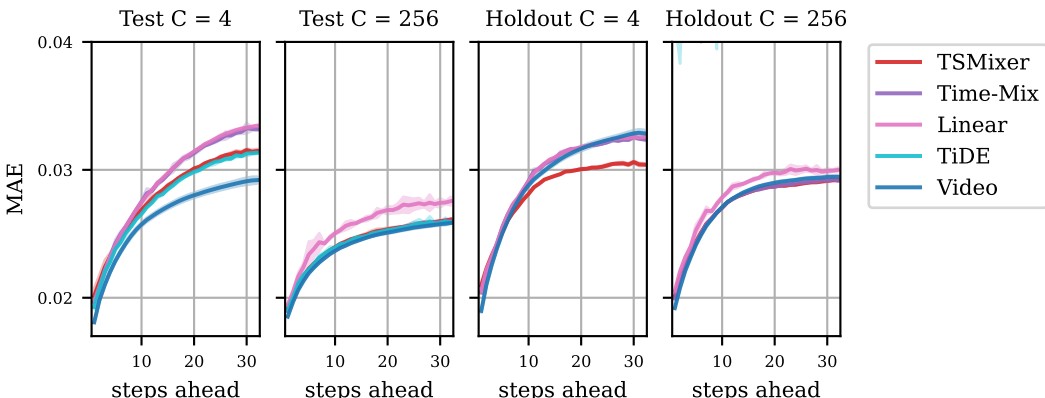

Figure 10: Comparison of volumetric video model with four trace-based models from ZAP-Bench (Anonymous, 2024) for short (left) and long (right) context on the benchmark test set (averaged over eight conditions) and the experimental condition held out from the training data. In remaining figures, we report performance relative to TSMixer and Time-Mix (a univariate MLP) for short and long context, respectively. Note that MAEs of TiDE on the holdout are higher than the axis limits, which is due to its reliance on stimulus covariates. We report the mean and two standard errors.

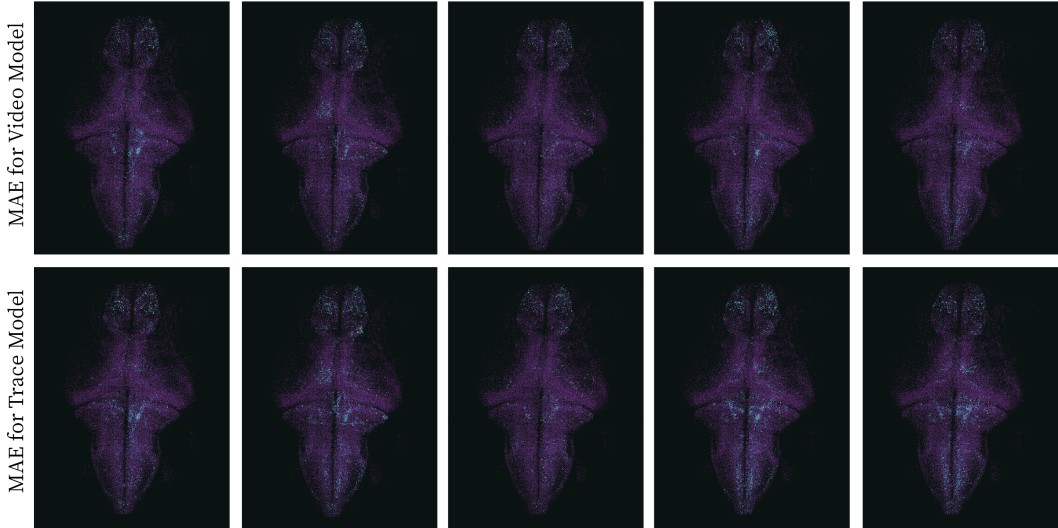

Figure 11: Illustration of MAE differences. Top row shows the MAE between target and predicted activity for a video model on five test set frames for the gain condition, $C = 4$, at 32 steps predicted ahead, with brighter colors indicating higher error. Bottom row shows corresponding MAEs on these frames for a trace-based model. When MAEs are averaged across all test set frames and neurons for this condition, the MAE difference between these models is approximately 0.005.

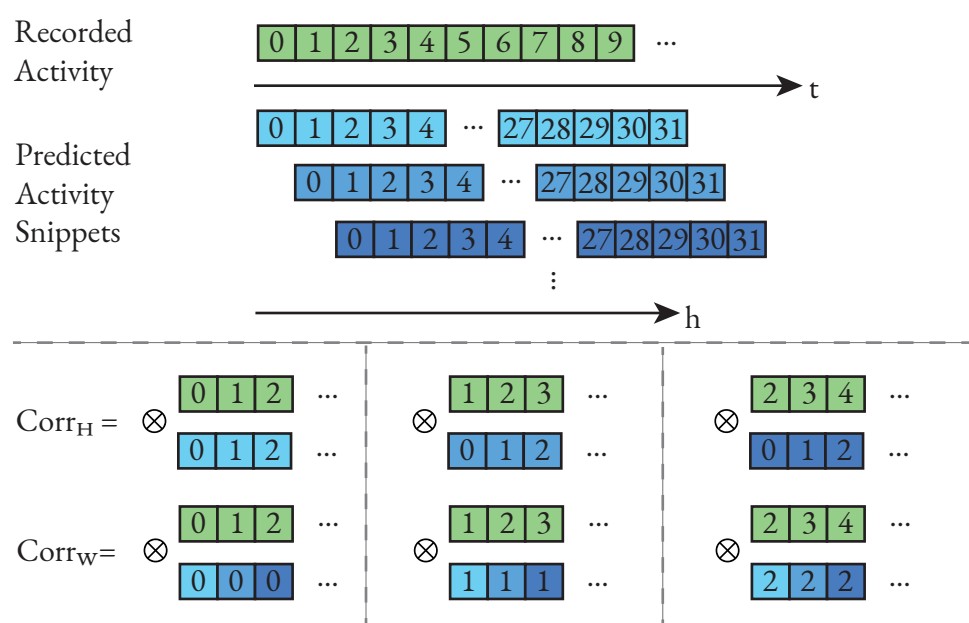

Figure 12: Illustration of the two types of correlation metrics (for a single neuron). Top: actually recorded activity (green) is aligned in experiment time $t$ with predicted snippets (blue) of activity of length $H = 32$ starting from various offsets. Bottom: correlations are always computed over 32 time steps between predicted activity and corresponding real recording. In $\text{Corr}_H$, complete predicted snippets are correlated with the recordings, and then averaged over starting points. In $\text{Corr}_W$, snippets are assembled from predictions at a specific lead time $h$, and correlated with the corresponding recordings. Reported metrics are averaged over all neurons.

### A.4 CORRELATION METRICS

Table 3: Test set $\text{Corr}_H \pm 2$ SE.

| Context | Video | Trace |
|---|---|---|
| $C = 4$ | $\mathbf{0.1511 \pm 0.0021}$ | $0.1080 \pm 0.0090$ |
| $C = 256$ | $\mathbf{0.1874 \pm 0.0004}$ | $0.1650 \pm 0.0022$ |

To better measure the quality of the temporal structures predicted by the model, we also computed two types of correlation metrics $\text{Corr}_W$ and $\text{Corr}_H$, which compare recorded and predicted activity over $H = 32$ steps, with the predictions assembled at constant lead time $h$ or from a specific starting point $t$, respectively (see Figure 12).

The correlation metrics paint a picture broadly consistent with that shown by the MAE, except in the long-context regime $C = 256$ where the video model outperfoms the trace-based models (see Table 3, Figure 13).

### A.5 COMPUTATIONAL COST ESTIMATES

The loss ablation in Figure 4 required around 5k GPU hours, pre-training and fine-tuning as shown in Sec. 4.1.2 around 14k GPU hours, comparing spatial to temporal context in Sec. 4.1.1 around 50k GPU hours, and the final results including the ablation on input resolution another 30k GPU hours. This makes a total of roughly 100k GPU hours used for the experiments presented in the paper.

A single training run of the best performing video model for $C = 4$ required $36\,\text{h}$, whereas the model for $C = 256$ required $120\,\text{h}$, both using 16 A100 GPUs. This compute cost is two to three orders

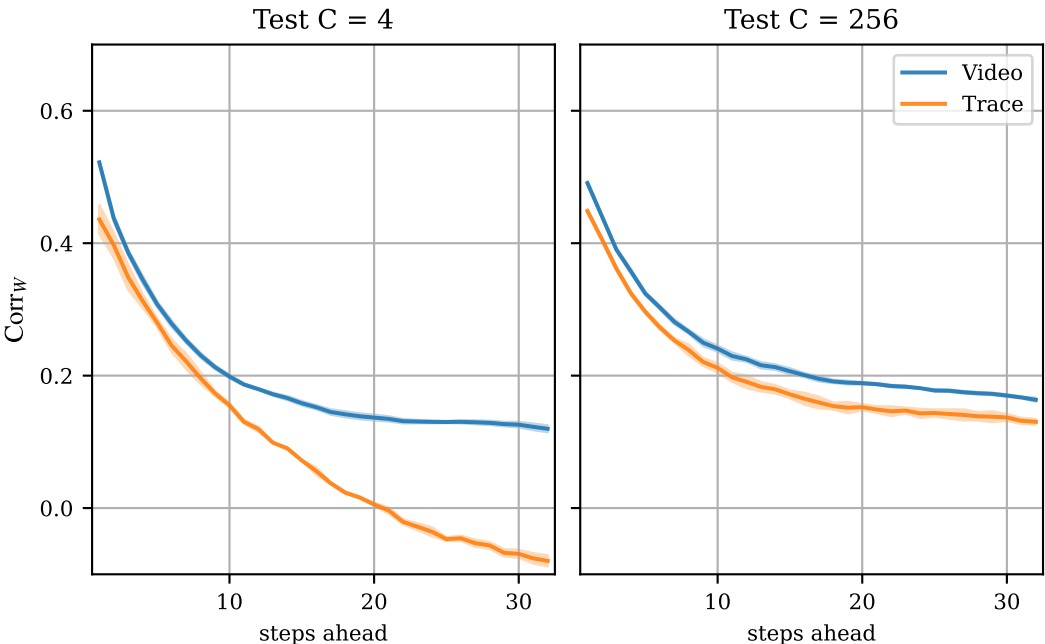

Figure 13: Comparison of volumetric video model with best-performing trace-based model in terms of $\mathrm{Corr}_W$ for short (left) and long (right) context on the benchmark test set (averaged over eight conditions), higher is better. We report the mean and two standard errors.

of magnitude higher than that incurred by training the baseline trace-based models, which require about $2\,\mathrm{h}$ on a single A100 GPU. However, video models require less raw data preprocessing relative to time series models, partially offsetting the increased cost.

