# OpenReview forum: "Forecasting Whole-Brain Neural Activity from Volumetric Video"
_ICLR.cc/2025/Conference — Submitted to ICLR 2025_

### Official Review · Reviewer_1oJD · 2024-10-24

**Soundness:** 2
**Presentation:** 2
**Contribution:** 3
**Rating:** 5
**Confidence:** 4

**Summary:**

This manuscript proposes the utilization of deep learning techniques for the prediction of neuronal activity recordings with fluorescent calcium, asserting superior performance over previous baselines. A series of ablation studies have revealed practical insights into model pre-training and hyperparameter tuning.

**Strengths:**

1.This paper employs a deep learning approach based on U-Net to directly process 4D neural activity recordings, circumventing complex preprocessing methods that may introduce performance degradation.

2.A series of ablation studies have revealed practical insights into model pre-training and hyperparameter tuning.

After reading the authors' rebuttal and the comments from Reviewer 5NDV, I agree that neural response prediction/forecasting directly from volumetric video is novel. This approach allows for minimal preprocessing of the data, which can be advantageous for deep learning-based methods.

**Weaknesses:**

1.The authors fail to elucidate the significance of this study anywhere within the main text, which raises questions about the necessity of forecasting whole-brain neuronal activity. The authors are encouraged to provide additional context in the abstract or introduction section.

2.This paper primarily utilizes the initial frames of neuronal activity recordings with fluorescent calcium indicators to predict subsequent frames, representing an application of U-Net in a specific domain. It does not offer additional insights or novel perspectives to advance the field of artificial intelligence. Therefore, this manuscript would be more appropriately submitted to conferences or journals focused on neuroscience or medical image processing, as it does not align with the thematic scope of ICLR.

3.The paper presents a limited comparison with only one baseline, namely the "trace-based model," as shown in Figure 6. It raises the question whether ZAPBench, as a benchmark, evaluated only this single model type. The authors are encouraged to include additional baselines for comparison to substantiate the superiority of the proposed method.

**Questions:**

1.On line 308, the authors refer to a "segmentation mask," yet on line 68, they claim that their method can directly process 4D data. The authors are requested to clarify the apparent contradiction in their narrative.

2.In the fourth contribution (line 108), the authors state that their proposed method is the "only approach that consistently benefits from multivariate information." However, I did not encounter any experimental justification related to multivariate information in the main text. If such experiments were conducted, please direct me to the relevant sections within the paper.


3.The description of temporal dimension processing on line 200 is unclear. I would like to confirm whether the authors' approach involves merging the temporal dimension with the batch dimension, such as transforming the data shape as follows: (batch, 2048, 1152, 72, T) --> (batch*T, 2048, 1152, 72). If not, please provide clarification on their methodology.

---

> ### Author Response · Authors · 2024-11-20
>
> Thank you for the review.
>
> > It does not offer additional insights or novel perspectives to advance the field of artificial intelligence. Therefore, this manuscript would be more appropriately submitted to conferences or journals focused on neuroscience or medical image processing, as it does not align with the thematic scope of ICLR.
>
> We respectfully disagree with this statement and refer the reviewer to the ICLR 2025 call for papers that lists applications to neuroscience as one of the relevant topics. Accordingly, we selected “Applications to neuroscience & cognitive science” as the primary area for this submission, as can be seen on top. We further emphasize that our work does indeed advance the field of AI by extending existing methods to a new, complex, and data-rich application domain, as well as by technical advances needed to scale them to volumetric videos.
>
> > The authors fail to elucidate the significance of this study anywhere within the main text, which raises questions about the necessity of forecasting whole-brain neuronal activity. The authors are encouraged to provide additional context in the abstract or introduction section.
>
> Thank you for this suggestion. We extended the introduction section of the paper to clarify the relevance of brain activity forecasting.
>
> > The paper presents a limited comparison with only one baseline, namely the "trace-based model," as shown in Figure 6. It raises the question whether ZAPBench, as a benchmark, evaluated only this single model type. The authors are encouraged to include additional baselines for comparison to substantiate the superiority of the proposed method.
>
> We compare to the best trace-based model on ZAPBench (c.f. line 383 in the revised manuscript), selected from a set of four state-of-the-art time series models. Also note that we are the first to propose a video-based model for such data and therefore unfortunately no existing video baselines to compare against exist. It is highly non-trivial to scale existing video-based models to the volumetric case while maintaining reasonable computational performance.
>
> > On line 308, the authors refer to a "segmentation mask," yet on line 68, they claim that their method can directly process 4D data. The authors are requested to clarify the apparent contradiction in their narrative.
>
> Our method performs video-to-video forecasting of neural recordings. However, to make a fair comparison and quantify the benefit of our method, we compared it to trace-based models. We do this by applying the segmentation mask to the voxel-level video forecast (i.e., model output).
>
>
> > In the fourth contribution (line 108), the authors state that their proposed method is the "only approach that consistently benefits from multivariate information." However, I did not encounter any experimental justification related to multivariate information in the main text. If such experiments were conducted, please direct me to the relevant sections within the paper.
>
> The best time series-based method on ZAPBench is a univariate model, which we also compare to. The multivariate time-series methods did not show a performance improvement. However, the video-based model performs best and relies on multivariate information (the model processes information from multiple cells simultaneously). Sec. 3.2 and Figure 5 in the paper clearly show and quantify the benefit of increasing the receptive field of the UNet, which corresponds to utilizing more multivariate information. Furthermore, based on a suggestion by reviewer 5NDV, we now also include an ablation evaluating the importance of the information contained outside of the cell masks. This additional experiment reveals that unsegmented voxels do not have a significant impact on the video model's performance.
>
> > The description of temporal dimension processing on line 200 is unclear. I would like to confirm whether the authors' approach involves merging the temporal dimension with the batch dimension, such as transforming the data shape as follows: (batch, 2048, 1152, 72, T) --> (batch*T, 2048, 1152, 72). If not, please provide clarification on their methodology.
>
> We do not merge the batch and temporal dimension but, as pointed out in lines 215 and following, use the temporal dimension as channels/features. Since we have a grayscale image, we would normally have a single channel at the input level. Here, we instead use the frames as input channels, which helps significantly with model scalability.
>
> We respectfully ask the reviewer to reconsider their evaluation in light of these clarifications.

---

> > ### Comment · Reviewer_1oJD · 2024-11-24
> > **Official Comment by Reviewer 1oJD**
> >
> > Thank you for your response . While some of my concerns have been addressed, I still have a few remaining questions.
> >
> > 1. I tentatively agree with the authors' perspective about the thematic scope of ICLR. However, I request that the authors provide references to previously published works on similar topics in ICLR as supporting evidence.
> >
> > 2. The motivation for this study is unclear. Although the authors discussed the significance of this study in the introduction, they did not address the necessity of forecasting whole-brain neuronal activity. **My primary concern is understanding the role of forecasting whole-brain neuronal activity in downstream tasks within neuroscience.**  If the authors' work is merely predicting subsequent frames of brain imaging data based on earlier frames, its contribution appears limited, as numerous similar studies already exist, such as video frame prediction in computer vision and time-series forecasting in machine learning.
> >
> > 3. Although the authors have added baselines, they remain insufficient. Furthermore, the study lacks a broader range of evaluation metrics; I note that only MAE is used throughout the manuscript. I would prefer to see evidence that the proposed large-scale training approach can improve the performance of certain meaningful downstream tasks.
> >
> > 4. I kindly request that the authors highlight the modified content in the manuscript during the rebuttal phase. Reviewing the changes by comparing the old and new versions line by line is very time-consuming.

---

> > > ### Author Response · Authors · 2024-11-25
> > >
> > > Thank you for your answer. We address your remaining questions below.
> > >
> > > > I tentatively agree with the authors' perspective about the thematic scope of ICLR. However, I request that the authors provide references to previously published works on similar topics in ICLR as supporting evidence.
> > >
> > > We respectfully suggest that the scope of the conference as outlined in https://iclr.cc/Conferences/2025/CallForPapers is not a matter of _our perspective_ here.
> > > Please note that applications to neuroscience have been within ICLR’s thematic scope since its inception, see e.g., the Call for Papers of ICLR 2013: https://iclr.cc/archive/2013/call-for-papers.htm
> > >
> > > Analysis of brain activity is clearly within the realm of neuroscience, and there are decades of prior work based on various modalities (ephys, fMRI, calcium recordings, etc). Extending this line of inquiry, *high-resolution whole-brain* activity analysis is a novel topic enabled by recent experimental advances, and unfortunately but correspondingly, related prior work is sparse. For a sample ICLR paper covering topics similar to ours, we refer the reviewer to https://openreview.net/forum?id=CJzi3dRlJE-, which provides a model of brain activity in a much simpler model organism (_C. elegans_).
> > >
> > > > The motivation for this study is unclear. Although the authors discussed the significance of this study in the introduction, they did not address the necessity of forecasting whole-brain neuronal activity. My primary concern is understanding the role of forecasting whole-brain neuronal activity in downstream tasks within neuroscience. If the authors' work is merely predicting subsequent frames of brain imaging data based on earlier frames, its contribution appears limited, as numerous similar studies already exist, such as video frame prediction in computer vision and time-series forecasting in machine learning.
> > >
> > > There is indeed extensive prior work in natural video prediction and time-series forecasting in various contexts. However, we argue that the nature of the application domain and the input data matter a lot, present different problems, and demand specialized solutions. In the video domain, our data is higher dimensional (3d+t for brain calcium movies vs 2d+t for natural videos) and generated by a small but complex system (larval zebrafish brain). Natural video models need to deal with lighting, reflections, perspective, occlusions, and movement. This is completely different in our case, where none of these are relevant and we are instead concerned with the spatiotemporal dependencies between the units of a complex network (neurons). In the paper we show that using the video representation leads to more accurate models than using time series.
> > >
> > > We view our work as fundamental research in advancing our ability to model and understand a complex dynamical system that is the brain. This can be seen as related to the area of brain simulation (see e.g.: [doi: 10.1016/j.cell.2015.09.029](https://pubmed.ncbi.nlm.nih.gov/26451489/)), which focuses on building detailed models of specific brain regions or circuits, incorporating knowledge of neuronal connectivity and biophysical properties. Our approach is complementary: it is more data-driven, able to model details which are not mechanistically understood, and rigorous in evaluation, in that model predictions are directly compared to recordings of a real brain.
> > >
> > > While at this stage of the research downstream applications are not a direct motivation, we sketch out here some potential directions for which our present work could be a foundation:
> > > - **Experimental design optimization**: Real recording sessions are in practice limited to about 2h, so the ability to predict responses to stimuli and perturbations in silico could be used to optimize what gets tested in the real world.
> > > - **Brain-computer interfaces (BCIs)**: Being able to precisely forecast neural activity could make it possible to build more efficient BCIs (lower error rates, lower latency).
> > > - **Anomaly detection**: Identifying deviations from typical patterns of activity could serve as biomarkers for neurological conditions and, in the future, for prediction of individual responses to therapy.
> > >
> > > > Although the authors have added baselines, they remain insufficient. Furthermore, the study lacks a broader range of evaluation metrics; I note that only MAE is used throughout the manuscript. I would prefer to see evidence that the proposed large-scale training approach can improve the performance of certain meaningful downstream tasks.
> > >
> > > Please see our response above regarding downstream tasks. Regarding metrics, we follow the setup of ZAPBench itself, which defined the dataset and the metrics that should be used to evaluate forecasting models built upon it.

---

> > > > ### Author Response · Authors · 2024-11-25
> > > > **(continuation of response above)**
> > > >
> > > > > I kindly request that the authors highlight the modified content in the manuscript during the rebuttal phase. Reviewing the changes by comparing the old and new versions line by line is very time-consuming.
> > > >
> > > > Please note that OpenReview already provides a PDF diff tool to highlight changes between revisions (see https://draftable.com/compare/HtOiPciNxaoO for the current paper). We also summarized all changes made to the manuscript in a top-level comment (https://openreview.net/forum?id=4UXIGATUTj&noteId=BF1AuZOTf5)

---

> > > > > ### Comment · Reviewer_1oJD · 2024-11-26
> > > > > **Official Comment by Reviewer 1oJD**
> > > > >
> > > > > Thank you for your response. Regarding the concern about whether this paper exceeds the thematic scope of ICLR, the authors have provided relevant references as evidence, which have addressed the issue satisfactorily. Additionally, I appreciate the authors' revisions to the abstract and introduction sections. As a result, I will increase the Presentation score to 2.
> > > > >
> > > > > However, in terms of experiments, the selected baselines and evaluation metrics remain insufficiently robust. Furthermore, as shown in Figure 6, under the conditions of Test C = 256 and Holdout C = 256, the proposed method shows almost no improvement over the baseline (Trace). Under the condition of Holdout C = 4, the performance of the proposed method is notably weaker than the baseline (Trace) when the number of steps exceeds 5. Therefore, I do not believe this paper meets the acceptance standard, and I will maintain my current score.

---

> > > > > > ### Author Response · Authors · 2024-11-27
> > > > > >
> > > > > > Thank you for raising the presentation score after our updates to the paper. We just wanted to note that as a result of discussion with reviewer 5NDV we have now extended the metrics to cover correlation scores, which paint a broadly similar picture to that presented by MAE. Measured this way, the video approach outperforms trace-based methods even in the long-context regime ($C=256$). We believe that additional non-video baselines are out of scope of this work and already sufficiently covered by ZAPBench itself.

---

### Official Review · Reviewer_5NDV · 2024-11-01

**Soundness:** 2
**Presentation:** 3
**Contribution:** 3
**Rating:** 6
**Confidence:** 4

**Summary:**

This paper proposes a new approach to neuronal response modeling by predicting/forecasting the volumetric video instead of the per-neuron calcium trace (dF/F) or spike train, which is the norm in neural response prediction. This approach allows the model to take advantage of the inter-cell activity and spatial organization of the population that is typically discarded when deconvolving the volumetric video to individual response traces. The authors evaluated a range of video and trace-based models on ZAPBench and showed that the video-based model outperforms trace-based models in short temporal context length conditions.

**Strengths:**

- To my knowledge, neural response prediction/forecasting directly from volumetric video is novel. This allows minimal preprocessing of the data which can be beneficial to deep learning-based methods.
- A wide range of training and evaluation conditions are compared, including trade-offs of spatial and temporal resolution, pre-training vs direct training, and training set size and combinations. These empirical results can guide future work in modeling neural responses.

**Weaknesses:**

Please find my suggestions for the following points in the Question section.
- A key motivation of this work, as stated by the authors in Section 2, is that the typical deconvolution step to convert volumetric video to dF/F response traces can lead to loss of information, such as cell spatial organization, inter-cell activities, etc. However, while the video-based model appears to outperform the trace-based model in short temporal context-length conditions (though similar performance in longer context length), it is unclear whether or not this is due to the additional information that exists in the raw data and that the video-based model is indeed taking advantage of such information.
- MAE might not be the most intuitive metric for getting a sense of how the (video and traced-based) models are performing. For instance, I am not sure if an MAE value of 0.02 is good or bad, or how big of a difference is an MAE of  0.02 to 0.04? In particular, I believe ZAPBench is a new dataset and we don’t have any other models to compare against these MAE values, other than the single trace-based model provided.
- Unclear trade-off in computation cost between video-based and trace-based models.

**Questions:**

Major
- It would be nice to test whether or not the trace-base model performed (relatively) poorly due to the lack of inter-cell activities or imperfect masking as suggested in Section 2 and Figure 2. I suggest applying the segmentation masks to the video, i.e. set regions outside of the identified cells as background, and train the video-based model on the masked videos. This should give us a sense of the influence of inter-cell activities and imperfect masking, and isolate the influence of spatial organization of the cells.
- I suggest the authors include metrics that are commonly used in neural response prediction so that readers can have a sense of how well these models are performing, such as normalized correlation ($CC_\text{norm}$) [1], or fraction of explainable variance explained (FEV) [2], basically metrics that takes trial-to-trial variability into account.
- Can the authors comment on the computational cost of the models? The authors stated in the hyperparameter search section and appendix A.3 that 16 A100 40GB GPUs are used to train the video-based model, and ~5k GPU hours (so ~300 hours in wall-time) was used in the loss ablation experiment in Figure 4, which is a considerable amount of computational time and cost. Can the authors share the time it took to train the final (best) video-based and trace-based models? I believe the authors should discuss the trade-off between the two approaches in computation cost if they are indeed substantially different. To clarify, I think it is fine for the method to be more computationally expensive than other methods, but it is important to point it out.

Minor
- What is the frame rate of the video?
- Why and how are the two temporal context lengths (4 and 256) selected? Does it make sense to predict the future 32 frames from only 4 frames?
- In the hyperparameters section and Figure 1, it is stated that the models optimize the trace-based MAE. Does this include the video-based model? Since the video-based model inputs and outputs a video, does it make a difference to optimize the recorded and predicted video MAE?
- How are the hyperparameters selected? Hand-picked or via some form of hyperparameter search (random search, bayesian search, etc.)

[1] Schoppe, Oliver, et al. "Measuring the performance of neural models." Frontiers in computational neuroscience 10 (2016): 10.

[2] Cadena, Santiago A., et al. "Deep convolutional models improve predictions of macaque V1 responses to natural images." PLoS computational biology 15.4 (2019): e1006897.

---

> ### Author Response · Authors · 2024-11-20
>
> Thank you for the review and for acknowledging the novelty of directly modeling volumetric video for forecasting brain activity and the extensive experimental evidence we presented.
>
> > “it is unclear whether or not this is due to the additional information that exists in the raw data and that the video-based model is indeed taking advantage of such information.”... “I suggest applying the segmentation masks to the video, i.e. set regions outside of the identified cells as background, and train the video-based model on the masked videos.”
>
> Thank you for this suggestion. We have now performed this experiment and added the results in the updated revision of the paper, finding that: “The grand average test MAE for that model (0.0266±0.0046) was not significantly different from that of the video model processing the complete volume (0.0267±0.0042). This indicates that the unsegmented voxels are unlikely to contain information that could improve forecasts and that any gains relative to the trace-based models can be attributed to the utilization of the spatial distribution of the underlying calcium signals within and across the segmented cells.”
>
>
> > “MAE might not be the most intuitive metric for getting a sense of performance” … “I suggest the authors include metrics that are commonly used in neural response prediction”.
>
> Thank you for the suggested metrics. Unfortunately, we do not have sufficient repetitions within the trials to reliably estimate the variances required by these metrics. Nonetheless, we updated the paper to refer to the mentioned prior work as potential future evaluation metrics for cases where more trials are available.
>
> While an improved MAE on the test set clearly indicates a better generalizing model, we agree that the absolute numbers are hard to interpret. To provide an intuition of what an MAE difference of e.g. 0.005 as reported in our results can look like, we included a new supplementary figure (Figure 10 in the revised manuscript).
>
>
> > “Can the authors share the time it took to train the final (best) video-based and trace-based models?”
>
> We have added this information to Appendix A.4. We acknowledge the significantly increased computational cost of this approach, but also note that the reduced extent to which the raw data needs to be preprocessed when forecasting directly in the video domain.
>
> > Minor questions
>
> - Minor Q1: The frame rate of the video is roughly 1 Hz.
> - Minor Q2: We use $C=4$ and $C=256$ as two extreme cases to assess the relevance of temporal context and quantify it in Figure 5. Indeed, it shows the difficulty of predicting $H=32$ frames from only 4 conditioning frames. However, generative natural video models can generate many more frames starting from only a few and we therefore argue that it does make sense to at least assess performance in this regime.
> - Minor Q3: We indeed optimize the trace-based MAE for the video models so that a fair comparison with the trace-based models can be made. If we optimize the voxel-wise MAE, the models perform relatively worse when evaluated with trace-MAE, as it corresponds to a different weighting of neurons by their size.
> - Minor Q4: The hyperparameters (learning rate, optimizer, weight decay, dropout) were optimized on the validation set using small grids of 3-4 values each, starting from commonly used values.

---

> > ### Comment · Reviewer_5NDV · 2024-11-27
> >
> > I thank the authors for their detailed response, please find my follow-up questions and comments below.
> >
> > > Thank you for this suggestion. We have now performed this experiment and added the results in the updated revision of the paper, finding that: “The grand average test MAE for that model (0.0266±0.0046) was not significantly different from that of the video model processing the complete volume (0.0267±0.0042). This indicates that the unsegmented voxels are unlikely to contain information that could improve forecasts and that any gains relative to the trace-based models can be attributed to the utilization of the spatial distribution of the underlying calcium signals within and across the segmented cells.”
> >
> > I thank the authors for performing this additional experiment. The model trained on segmented video should have some of the constraints as the trace-based model does (i.e. imperfect segmentation, elimination of inter-cell activities, etc), though it still achieved similar performance as the model trained on raw volumetric video. Does this result contradict some of the motivations stated in the paper: "While this is a natural choice, it loses information related to cell size, position and spatial distribution of intensities within it, and completely discards voxels that are not part of any segmentation mask or incorrectly segmented. Figure 2 depicts these potential issues."?
> >
> > > Thank you for the suggested metrics. Unfortunately, we do not have sufficient repetitions within the trials to reliably estimate the variances required by these metrics. Nonetheless, we updated the paper to refer to the mentioned prior work as potential future evaluation metrics for cases where more trials are available. While an improved MAE on the test set clearly indicates a better generalizing model, we agree that the absolute numbers are hard to interpret. To provide an intuition of what an MAE difference of e.g. 0.005 as reported in our results can look like, we included a new supplementary figure (Figure 10 in the revised manuscript).
> >
> > If there aren't enough repetitions in each trial to use metrics that account for trial-to-trial variability, can the authors at least provide the single-trial correlation between recorded and predicted responses (average over neurons)? Again, it is hard to interpret MAE values, especially without knowing the range of the calcium response.
> >
> > > We have added this information to Appendix A.4. We acknowledge the significantly increased computational cost of this approach, but also note that the reduced extent to which the raw data needs to be preprocessed when forecasting directly in the video domain.
> >
> > Thank you for providing the computational cost information. Given the vast difference in computation cost between the trace-based and video-based models (2 hours on a single A100 GPU vs 36 hours on 16 A100 GPUs), I encourage the authors to include this limitation in the main text, perhaps in the discussion/conclusion section.
> >
> > All in all, my main concerns are:
> > 1. It remains unclear why training the model on raw volumetric video is more advantageous than the existing approach of segmented calcium traces. The additional analysis provided by the authors shows that the models trained on raw and segmented volumetric video achieved similar performance, so is the performance gain solely due to the additional information on the spatial organization of the cells? If so, the experiments conducted in this paper haven't demonstrated that.
> > 2. MAE can be a good optimization objective, but it is unintuitive as a metric for comparing models. I suggest adding correlation-based metrics (single trial correlation, average correlation, FEV, etc.), which are commonly used in neural response prediction [1, 2].
> >
> > For these reasons, I maintain my original score.
> >
> > [1] Schoppe, Oliver, et al. "Measuring the performance of neural models." Frontiers in computational neuroscience 10 (2016): 10.
> >
> > [2] Turishcheva, Polina, et al. "The dynamic sensorium competition for predicting large-scale mouse visual cortex activity from videos." ArXiv (2023).

---

> > > ### Author Response · Authors · 2024-11-27
> > >
> > > Thank you for the additional comments!
> > > > I thank the authors for performing this additional experiment. The model trained on segmented video should have some of the constraints as the trace-based model does (i.e. imperfect segmentation, elimination of inter-cell activities, etc), though it still achieved similar performance as the model trained on raw volumetric video. Does this result contradict some of the motivations stated in the paper: "While this is a natural choice, it loses information related to cell size, position and spatial distribution of intensities within it, and completely discards voxels that are not part of any segmentation mask or incorrectly segmented. Figure 2 depicts these potential issues."?
> > >
> > > Indeed, we believe that this result narrows down the range of possible sources of the empirically improved performance and clarifies which hypotheses we posed as the motivation for this line of work in section 2 are correct. Specifically, it suggests that segmentation quality is not a significant limitation, and that the gains can be attributed to the better utilization of the spatial distribution of the observed fluorescence signal. This interpretation is further supported by the spatial context results in Fig. 5 and the input resolution results in Table 2, suggesting that the correlations between cells, but not the intra-cellular distribution of fluorescence, drive the increased accuracy. We included these comments in section 4.2 of the paper.
> > >
> > > > If there aren't enough repetitions in each trial to use metrics that account for trial-to-trial variability, can the authors at least provide the single-trial correlation between recorded and predicted responses (average over neurons)? Again, it is hard to interpret MAE values, especially without knowing the range of the calcium response.
> > >
> > > Thank you for this suggestion. We have added correlation scores between predicted and recorded responses to the paper (Figure 13 and Table 3 in appendix A.4), in addition to mentioning the range of the underlying df/f calcium signals (-0.25, 1.5) in section 2. We find that the proposed video-based model achieves a stronger correlation with the recorded responses, 40% higher in the short context regime ($C=4$) and 14% higher in the long context regime ($C=256$). Furthermore, we observe that for $C=4$, video models show positive correlation throughout the complete prediction window $H=32$, unlike trace-base models which become uncorrelated after 20 time steps.
> > >
> > > > Thank you for providing the computational cost information. Given the vast difference in computation cost between the trace-based and video-based models (2 hours on a single A100 GPU vs 36 hours on 16 A100 GPUs), I encourage the authors to include this limitation in the main text, perhaps in the discussion/conclusion section.
> > >
> > > We have now added a mention of this in the discussion section.
> > >
> > > > It remains unclear why training the model on raw volumetric video is more advantageous than the existing approach of segmented calcium traces. The additional analysis provided by the authors shows that the models trained on raw and segmented volumetric video achieved similar performance, so is the performance gain solely due to the additional information on the spatial organization of the cells? If so, the experiments conducted in this paper haven't demonstrated that.
> > >
> > > We observe that once the signal from the unsegmented voxels is eliminated, as was the case in the experiment we ran, the only advantage that the video model has relative to the trace model is access to the spatial distribution of the underlying fluorescence signals. Together with the network FOV and input resolution results already reported in the paper, we believe these data points to be strongly indicative of the video model better utilizing the multivariate nature of the input signals.
> > >
> > > > MAE can be a good optimization objective, but it is unintuitive as a metric for comparing models. I suggest adding correlation-based metrics (single trial correlation, average correlation, FEV, etc.), which are commonly used in neural response prediction [1, 2].
> > >
> > > This has now been addressed. We hope the reviewer and other readers will find that these additional metrics provide more context for the interpretation of our results.

---

> > > > ### Author Response · Authors · 2024-12-02
> > > >
> > > > One additional comment from our side. After the discussion above, we decided to perform additional checks to narrow down the sources of improved performance. We rendered two "synthetic calcium movies": one ("rendered traces") with the voxels of the segmented cells set to the corresponding trace value (uniformly throughout each cell), and one ("shuffled traces") with the traces randomly reassigned to different cells. Training video models on this data, we observed that the model using "rendered traces" performs equivalently to the ones using the full df/f volume and the segment-masked df/f volumes. The "shuffled traces" variant however showed statistically worse test-MAE.
> > > >
> > > > These additional experiments strengthen our original conclusion that the distribution of fluorescent signal within individual cells and throughout unsegmented voxels has negligible impact on forecasting accuracy, and that the additional accuracy of the video model stems for the utilization of multivariate, cross-cell information -- precisely the type of information that trace-based models in ZAPBench have difficulty using.
> > > >
> > > > We will add a discussion of these experiments to the camera-ready version of the paper should it be accepted.

---

> > > > > ### Comment · Reviewer_5NDV · 2024-12-02
> > > > >
> > > > > I thank the authors for the additional experiment on original segmented cells vs shuffled cells, I believe this is a nice test to isolate the performance gain due to the spatial organization in the volumetric video data. Also thank you for adding correlation as a metric.
> > > > >
> > > > > I have updated my score accordingly.

---

### Official Review · Reviewer_NKSZ · 2024-11-03

**Soundness:** 4
**Presentation:** 3
**Contribution:** 3
**Rating:** 8
**Confidence:** 3

**Summary:**

This paper introduces a novel method for neural activity forecasting of zebrafish that works on the raw volumetric video instead of using standard preprocessing that reduces the original 4D space to 1D (trace matrix) and disregards spatial relationships between neurons. To do this, a u-net architecture is employed, taking advantage of a large scale neural dataset and performing extensive ablations for model selection. Multiple measures were taken to enable scaling the architecture for this computationally expensive problem, such as using the temporal context dimension as input channels, lead-time conditioning, and distributed training. The ablation results show that (1) pretraining on other specimens does not help, (2) there is a trade-off between spatial and temporal context, and (3) that downsampling input resolution up to 4x is beneficial to performance. Compared to the best trace-matrix models, the proposed multivariate model achieves 10% reduction in MAE for the short context forecasting setting in both the test and the holdout sets, while it is comparable to the trace-matrix models for long-context forecasting in the test set and 12% better in the holdout set.

**Strengths:**

The paper is original in the sense of creative combinations of existing ideas (the components of the model that enable scaling), application to a new domain (forecasting on minimally preprocessed / raw data from weather to neural data), as well as removing limitations from previous works (removing dependency on segmentation mask accuracy and avoiding loss of information caused by conversion to trace-matrix). Writing style is very high quality, and all the methods and results get across in a relatively clear manner to the reader. To the specific line of research (forecasting neural data), the method proposed seems to be a significant step forward in the field’s development, moving from hand-crafted to learnable features.

**Weaknesses:**

There are some presentation issues that impact the understanding of a reader that is not very familiar to forecasting neural data.
* These are mainly in the abstract and introduction, while once entering section 2 all misunderstandings of this reader were resolved. Nevertheless it would be beneficial for the paper to have them fixed. (1) Until section 2 it was not very clear that the goal is to predict future steps from previous steps, and not one neural modality (e.g. electrical signal) from another (e.g. blood oxygen). (2) It was not clear that the neuron segmentation mask is applied in both the trace models and the proposed model, but at different points in the pipeline, i.e. in the latter the forecasting itself is done on the volumetric video and afterwards the mask is applied before computing the error - without knowing this it is not clear how the two methods can be compared fairly. It would also help if in Figure 1 the same notation was used between orange (trace) and blue (proposed) in the segmentation mask block, i.e. instead of “Extract Neurons” and “Mask Neurons” say “Apply segmentation mask” in both cases. (3) In the abstract the phrase “we design a model to handle the high resolution and large receptive fields…” is structured in a confusing way where the reader does not understand if the large receptive fields are an aspect of the model or the recordings. (4) Minor - the footnote on page 2 is not so much footnote information, but rather more suitable for the main text.
* Additionally, Figure 2 should have a more informative caption that explains better what is shown, e.g. it is not clear what the colored blobs are, segmentation masks?
* Is H=32 the only setting that is tested and why? Not sufficiently described in the paper.
* In the conclusion, calling the findings counterintuitive seems excessive; there is no reason to assume that high input resolution, pretraining, or increased model capacity works well for all domains and applications. Results sufficiently showcase enough reasons why these sometimes useful settings might lead here to overfitting, distribution shifts, etc.

**Questions:**

Could "scaling the field of view of the network so the size remains constant while increasing to full resolution" make the full-resolution model handicapped in terms of field of view? There is a chance this is already answered in the paper but the reader missed it.
Other than that, fixing the presentation issues in the previous section.

---

> ### Author Response · Authors · 2024-11-20
>
> Thank you for the positive review acknowledging the quality of the work and the significance of advancing neuronal forecasting. In the following, we address your concerns and questions:
> - We edited the abstract and introduction to improve the clarity issues you pointed out regarding the task, segmentation mask, and receptive fields.
> - We extended the caption of Figure 2 as suggested. Indeed, the colored blobs denote the segmentation masks of individual cells.
> - In the paper we follow the benchmark setup of ZAPBench, which poses the task of predicting $H=32$ future timesteps. Longer time horizons currently seem to be less interesting as performance of all existing methods significantly deteriorates to the level of naive baselines beyond 32 timesteps. We note that shorter time horizons are automatically part of the current evaluation.
> - We completely agree with your assessment that these techniques should not be expected to work (out of the box) in all domains. This is what we wanted to express by calling them "counterintuitive" -- to call the attention of the reader to the fact that different strategies might be required in the video forecasting domain. We have now softened the statement and just say "contrary to our expectations" instead.
>
> > Could "scaling the field of view of the network so the size remains constant while increasing to full resolution" (line 349) make the full-resolution model handicapped in terms of field of view?
>
> The higher resolution models use one and two more downsampling blocks for the 2x and full resolution models, respectively. Therefore, we make sure they are not handicapped in terms of their capacity and field of view. Details can be found in Appendix 1.3 and specifically from line 644 starting with “To ensure equitable comparison, the architectures of these models were kept broadly consistent, with necessary adjustments to accommodate the differing input resolution”.

---

> > ### Comment · Reviewer_NKSZ · 2024-11-24
> >
> > Thank you for making the requested changes, and clarifying those points.

---

### Author Response · Authors · 2024-11-20

We thank all reviewers for their feedback, and appreciate the positive comments related to the novelty of  our approach of using deep learning techniques to directly process 4d neural activity recordings, the significance of this work for the field of neuroscience, and the extensive evaluations and ablations.
We replied to each reviewer individually to address their suggestions.

Limited baselines were listed as a concern by multiple reviewers, so here we additionally highlight that ZAPBench itself reports the results of multiple timeseries-based forecasting approaches, and we compare to the *best* performing methods for the $C=4$ and $C=256$ conditions (TSMixer, and MLP, respectively) for clarity. For completeness, we added Figure 9 to the appendix, which contains the other baselines.

The revised paper contains the following changes:
- Revised the abstract and introduction to improve clarity regarding the task, segmentation mask, receptive fields, and relevance of neural activity forecasting.
- Updated labels in figure 1.
- Updated caption of figure 2.
- Updated figures 6, 7, and 8 to reflect revised hold-out condition results on ZAPBench for trace-based models, along with corresponding text changes.
- Updated discussion of metrics with additional references.
- In the conclusion section, rephrased 'counterintuitive' to 'contrary to our expectations'.
- Added masking experiment in Appendix A.2.
- Added supplementary figure (Fig. 9) to Appendix A.3 showing performance relative to four trace-based approaches included in ZAPBench.
- Added supplementary figure (Fig. 10) to Appendix A.3 illustrating MAE differences.
- Added more detailed compute stats to Appendix A.4.

---

### Meta-Review · Area_Chair_F8j3 · 2024-12-22

**Metareview:**

The paper proposes a UNet-based approach for forecasting whole-brain neuronal activity using volumetric video data, achieving strong results on the ZAPBench benchmark. While the focus on leveraging raw volumetric video data is promising, the paper would benefit from engaging more with the broader field of 4D spatio-temporal modeling, as these approaches are commonly used in areas like fMRI analysis and climate forecasting. The authors do state in the rebuttal that there exist no video-based models for this dataset. However, since this work is strongly motivating its methodological innovation, comparison against an array of 4D spatio-temporal models needs to be compared for this new dataset. Even if the UNet is proposed as an option in this paper, this is mentioned as one of the baselines in the anonymous reference which introduces the ZAPBench benchmark, further questioning the technical contribution of this work. Therefore, although the application brings important contributions, the methodological contributions, the thrust of this work, need to be improved.

**Additional Comments On Reviewer Discussion:**

During the rebuttal period, reviewers requested additional baselines, and the authors responded by incorporating baselines from the anonymous reference (Anonymous, 2024), which is the ZAPBench benchmark paper. While the reviewers appeared satisfied with this response, it is notable that these baselines do not include comparisons with volumetric video models, a critical omission given the paper’s focus on volumetric forecasting. Additionally, the paper overly relies on the benchmark results of Anonymous, 2024, with UNet cited as a volumetric video model baseline but with additional details deferred to the referenced work, raising concerns about the independence of this submission. Although the reviewers raised their scores, they may not have fully recognized these issues. Given the standard nature of the technical contributions, the model-specific analyses, and the close dependence on the anonymous submission, I have decided to recommend rejection.

---

### Decision · Program_Chairs · 2025-01-22

Reject